# Strategies for Alleviating Spatial Disadvantage: A Systems Thinking Analysis and Plan of Action

**Richard Tucker** [1], **Louise Johnson** [2], **Jian Liang** [3,*] and **Steven Allender** [4]

1   HOME Research Hub, Deakin University, Geelong, VIC 3220, Australia
2   LCJ Research & HOME Research Hub, Deakin University, Geelong, VIC 3220, Australia
3   Department of Finance, Deakin Business School, Deakin University, Burwood, VIC 3125, Australia
4   Global Obesity Centre, Institute for Health Transformation, Deakin University, Geelong, VIC 3220, Australia
*   Correspondence: jerry.liang@deakin.edu.au

**Abstract:** Within Australian cities there is significant socioeconomic disparity between communities, which is an obstacle to sustainable urban development. There is a voluminous amount research into the causes and some of the ameliorative actions to address socio-spatial disadvantage, though many studies do not localize or systematize their analyses. This paper presents the results of a co-design process conducted with community stakeholders using innovative realist inquiry and system mapping to answer the question: what are the impacts and drivers of socioeconomic and spatial disadvantage in a regional city in Victoria, Australia, and what actions might ameliorate these in three localities? Participants identified 24 separate causes and impacts of acute socioeconomic disadvantage. Using system maps, these community members developed 13 intervention ideas for action with potential to positively impact health and wellbeing, education, housing, employment, and livability, and be translatable to policy positions. The paper therefore presents a unique method of enquiry into spatial disadvantage and a grounded set of strategies for positive action.

**Keywords:** disadvantage; socioeconomics; socio-spatial disadvantage; systems thinking; Australia

## 1. Introduction

Socio-spatial disadvantage has been a long-term issue for communities and those trying to analyze and ameliorate the many challenges associated with this status: be it long-term unemployment, poor health, food insecurity, or housing stress. Alleviating socio-spatial disadvantage is crucial to sustainable urban development. Research in this field is aligned with the United Nations Sustainable Development Goals, such as reducing regional inequality, enhancing the inclusiveness of society, and reducing poverty. While recognized as a multi-faceted problem needing diverse solutions, existing research and practice has clearly not been effective, suggesting the need for different approaches. Here we present a place-based systems approach which offers insights derived from community engagement: a critical assessment of existing practices and strategies which may well deliver more impactful outcomes as a consequence.

The 2016 SEIFA Index defines three Geelong suburbs—Corio, Norlane, and Whittington—as amongst the "most disadvantaged" in Victoria. Despite the many strengths of these diverse communities, concentrations of socioeconomic disadvantage in these communities present long-term and severe problems for many residents. In the project reported here, researchers worked with a local council and community stakeholders to provide recommendations to ease socioeconomic disadvantage in Geelong, a regional city of 244,000 in Victoria, Australia. Here, we present the results of the final stage of the project; a co-design process conducted with community stakeholders using realist inquiry and system mapping to answer the research question: what are the impacts and drivers of socioeconomic and spatial disadvantage in the three most disadvantaged suburbs in Geelong, and what actions might ameliorate these? The question was addressed in three stages, which aimed to:

(1) create a shared understanding of the impacts of acute socioeconomic disadvantage in place, (2) develop a set of practical ideas to address some of these impacts in the light of existing practice; and (3) identify which of these ideas should be prioritized.

A workshop was held early in 2022 with 18 community stakeholders on the impacts of acute socioeconomic disadvantage in place, comprising community representatives, staff working in related departments in the local council (the City of Greater Geelong (CoGG) *n* = 6), and representatives from state government, health, community services, and educational and not-for-profit organizations. In this we utilized a unique methodology—Systems Thinking in Community Knowledge Exchange (STICKE)—developed by Deakin University as a tool for accessing and analyzing complex systems. Discussions were used to create causal loop diagrams representing the causes and impacts of acute socioeconomic disadvantage in Geelong. These diagrams were used in a participatory process to identify existing actions to address the impacts of acute socioeconomic disadvantage, identify areas where more effort would be valuable in preventing or alleviating these impacts, and to describe and prioritize new actions to address these deficiencies based on feasibility and likely impact.

The workshop identified 24 separate causes and impacts of acute socioeconomic disadvantage. These were categorized into five themes (Policy (in relation to governmental funding for community and social infrastructure); Safety (in relation to neighborhood reputation and place-stigma); Housing (particularly access to affordable and appropriate housing); Employment and Work; and Social Networks, Support, and Exclusion), and related to five domains of local government practice (health and wellbeing, education, housing, employment, and livability). Using system maps, community members developed 13 intervention ideas for action by and/or with local government, including: the improvement of local, state, and federal governmental funding policy to alleviate disadvantage; improving neighborhood reputation (place-stigma); stipulating the increased supply of affordable and appropriate housing as a prerequisite of all development in the three communities; tailoring support to the needs of local small business owners to improve employment opportunities for residents; and empowering the voices of local communities.

As shall be discussed in detail, our findings demonstrate:

- that research outcomes must be discussed with key stakeholders responsible for policy development and service delivery to ensure that the research leads to action;
- how communities see their current context interacts with the evidence about the drivers of concentrated socioeconomic disadvantage and what the priority actions would be to address these causes and impacts; and
- how community stakeholders can identify and develop strategies to address drivers unique to their communities.

## 2. Background

### 2.1. Socioeconomic Disadvantage

Socioeconomic disadvantage takes many forms, including exclusion from quality and affordable housing and many everyday activities, recreational, educational, health, and other services and facilities. This can lead, for example, to poor employment, health, livability, and educational outcomes. Ten years ago, a systematic review was authored by Wiesel and Pawson [1] of Australian policy, practice, and literature on the processes that lead, and urban policy responses to, concentrations of disadvantage. They identified the structural causes of disadvantage as principally labor market, housing system and policy drivers. The work also identified the features of locations that negatively impact residents' lives as a combination of: (1) neighborhood effects (whereby living in a poor area compounds individual disadvantage), and (2) correlated neighborhood effects (physical factors that disadvantage areas due to spatial disadvantage, i.e., the housing stock and an area's location relative to employment and services).

It is worth here differentiating the terminology associated with this understanding. Disadvantage (or socioeconomic disadvantage) is an umbrella notion, which applies to

people rather than places, "embracing a series of concepts (including poverty, deprivation, social exclusion and social capital), which describe different aspects of distributive and/or social inequality" [2]. Spatial disadvantage is an "umbrella term incorporating three concepts: spatial disadvantage, places with social problems, and concentrated disadvantage, the latter referring to places accommodating a disproportionate number of socio-economically disadvantaged people" [2], such as those who experience poverty and/or, for example, disability, ageing, or single parenting [3]. Spatial disadvantage encompasses a range of deficits relating to socioeconomic status: the availability and quality of resources such as employment, services, transport, and amenity [4]; and levels of social 'dysfunction,' [3], such as crime and violence [5,6]. It is argued that the spatial concentration of disadvantaged people exacerbates their disadvantage [3], and can facilitate a 'culture of poverty' and dependence [7] characterized by lack of economic self-sufficiency, violence, drug dependency, and poor educational aspiration [8–10].

Foregrounding this research with community stakeholders, we completed systematic reviews of the research literature since 2010 (overlapping just over ten years with the Wiesel and Pawson 2012 review [1]) to understand prior work on identifying the socioeconomic factors driving outcomes for communities experiencing disadvantage, and on the intervention strategies used and evaluated to address disadvantage. Summaries of this evidence review were presented to the workshop participants to provide the research context for their deliberations. In this, efforts were taken to make the brief neutral by limiting and generalizing information in relation to the primary workshop research objective to identify strategies to alleviate spatial disadvantage. Our review of 78 papers on the socioeconomics of disadvantage identified 52 distinct socioeconomic factors driving outcomes for communities (in-press citation removed for anonymity). An evidence map representing the field of research shows the breadth of factors identified across research fields and an indication of their relative importance in terms of research coverage (Figure 1). The factors are categorized according to four contexts (economic, health, social, and urban environment) and 13 determinates commonly recognized in the well-cited literature [11,12].

By far the greatest number of studies (33) were in the Health and Wellbeing domain, with a further 11 including Health and Wellbeing in combination with other domains. Significantly, 15 studies spanned all domains, and a further 13 studies spanned at least two domains. Outside of Health and Wellbeing, only 17 studies focused on a single domain, with the majority of these looking at housing. The high number of studies spanning more than one domain reflects, as a recent productivity commission report on inequality in Australia puts it, the fact that "disadvantage is a multidimensional concept that can take the form of low economic resources (poverty), inability to afford the basic essentials of life (material deprivation) or being unable to participate economically and socially (social exclusion)" [13].

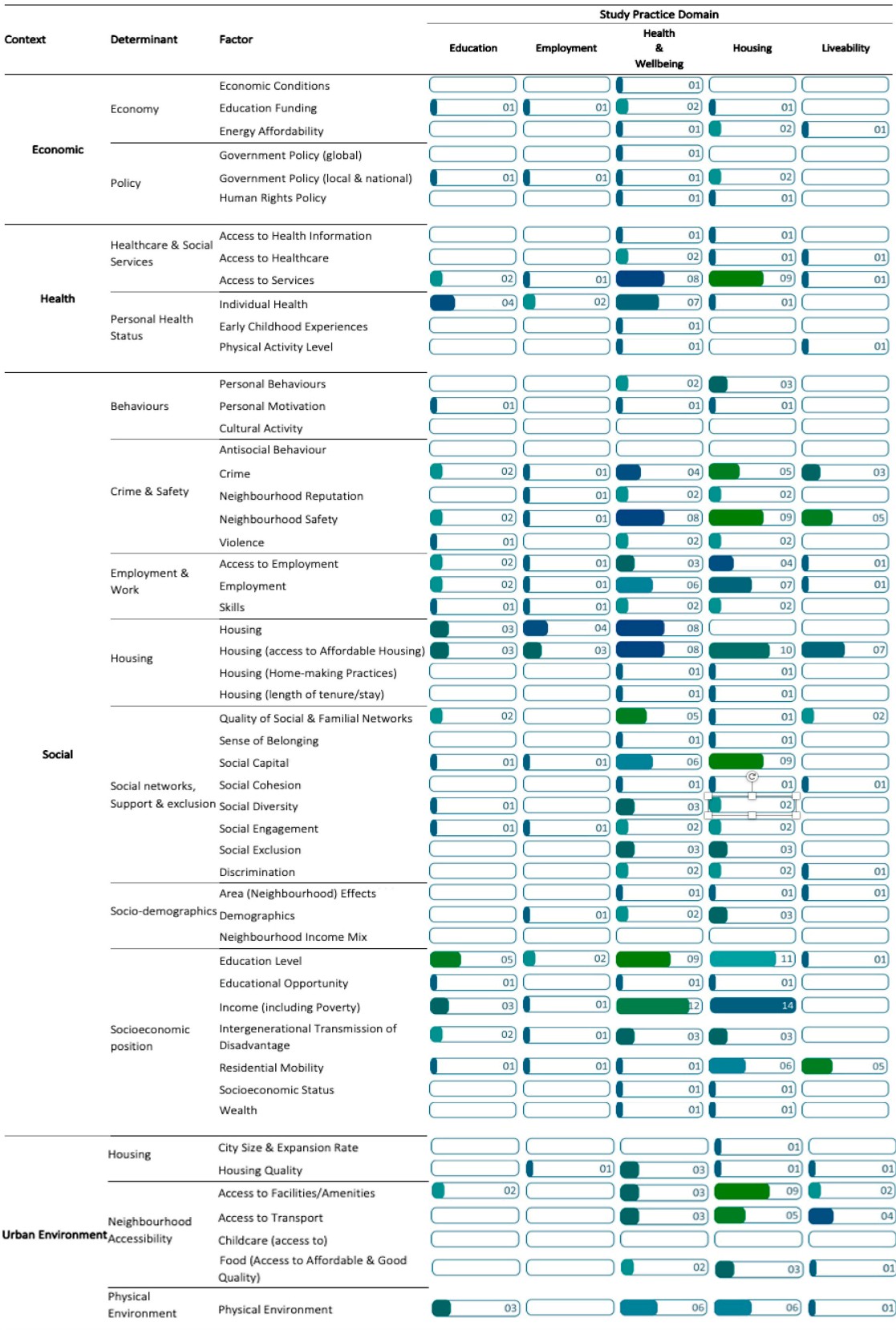

**Figure 1.** Evidence map of 52 distinct socioeconomic factors driving outcomes for communities experiencing disadvantage.

As spatial disadvantage is a wicked as well as intractable problem, in the peer reviewed international literature identified by our systematic review of intervention strategies, unsurprisingly around half of the interventions reported span practice domains, with nearly a quarter crossing all domains. Most interventions are focused on community engagement and development, food access initiatives, health policy and promotion, and neighborhood renewal. Many studies integrate health with housing as part of community development and neighborhood renewal. The spanning of disciplines here acknowledges the growing recognition of housing impacts on community health and that "vibrant neighborhoods are vital to health" [14]. The nature of intervention research makes it clear that spatial disadvantage is multidimensional, and requires multidimensional solutions—multi-pronged in focus, and drawing on cross-sector collaboration. For example, teams working on strategies integrating public health and housing policy saw this as being possibly able "to contribute to the 'triple win' of health and well-being, equity, and environmental sustainability" [15], and saw "strong potential for cross-sector collaborations to reduce health disparities and slow the growth of health care spending, while at the same time improving economic and social well-being" [14,16]. Initiatives often focus on community engagement towards community-informed types of renewal, with success here largely dependent on the quality of community engagement, and quality of partnerships and communication between stakeholders. Attree et al. caution, however, that community engagement might result in some unintentional risks to well-being, particularly for individuals with disabilities, including "exhaustion and stress, as involvement drained participants' energy levels as well as time and financial resources" [17].

When it comes to evaluating the impacts of place-based interventions, one size does not fit all, and the evaluation process must be context dependent [18,19]. While impact evaluation largely aims to capture the local effects of a program, knowing if results can be replicated is valuable. In other words, "given the very local nature of place-based strategies, understanding how impacts are achieved in one community can provide useful lessons when similar strategies are enlisted in other communities" [18]. Similarly, "there is insufficient evidence to determine whether one particular model of community engagement is more effective than any other" [20].

Finally, we made a comparison of the Australian experience of the last two years with pre-pandemic research. The bulk of the pre-pandemic inequality data came from the Inequality in Australia 2020 report [21–23], which was an analysis of demographic data identifying a widening gulf between people with the lowest and highest incomes. Our pandemic data came from the following sources: for education [24–30]; for employment [31–34]; for health and wellbeing [35–38]; for housing [39–44]; and for livability [45–47]. While we report in detail on this comparison elsewhere (in-press citation removed for anonymity), in summary we found that many determinants of spatial disadvantage remained during the pandemic: socioeconomic position, social networks, support and exclusion, access to health and social services, personal health, familial employment, housing affordability, and neighborhood accessibility. However, other determinants were accentuated: digital access (to education and telehealth), gender (principally impacts on women), age (particularly impacting the employment of young adults), and food security.

Existing academic literature therefore highlights the multifaceted nature of socio-spatial disadvantage and the need for a diverse response to it. However, it also highlighted the importance of local assessments and community engagement in the definition and development of ameliorative strategies, but also the limited success of such actions. More recent work has indicated that COVID-19 exacerbated but also added new intensities to elements of disadvantage. There is therefore a need to approach the problem in new ways, especially in the light of the pandemic experience.

### 2.2. Community-Based Systems Dynamics

Systems science, and specifically community-based systems dynamics [48], can provide techniques to understand inherent complexity from the point of those living in com-

munities experiencing concentrations of socioeconomic or other forms of disadvantage. Systems thinking and research techniques offers the possibility of unpacking, at a local level, those elements of any one system that those present assess as important to their understanding of the problem and to its amelioration. The subsequent understanding of the system includes the status of the community, resources, and political acceptability of change, among others [49]. Interventions built on these techniques are more suited to place than externally developed and non-consultative intervention strategies. These techniques have previously been used to support communities and cohorts in Geelong that experience disadvantage (references removed for anonymity).

In this paper we report on a systems thinking workshop, which set out to:

- Create a shared understanding of the drivers of concentrated disadvantage in Geelong;
- Provide an overview of existing literature to community members and help them understand how interventions may work for communities experiencing long-term concentrations of socioeconomic disadvantage in Geelong;
- Develop a set of practical ideas to address some of the impacts of these drivers;
- Identify policy settings and potential initiatives that would support these changes; and
- Identify points in the system where local government supported intervention could fruitfully be used.

### 3. Setting

Located 75 kms from Victoria's capital city, Melbourne, Geelong is the second city of the State with nearly a quarter of a million people. Situated north of the city center, the adjacent suburbs of Corio and Norlane are the former heartland of Geelong's once-thriving manufacturing industry. Corio has approximately 16,000 residents across 19 square kilometers, while Norlane has 8300 residents within its 5-square kilometer boundary. Whittington is south-east of the city center, with 3900 residents across 1.5 square kilometers [50].

In Australia, disadvantage is measured by two key instruments: the SEIFA index of Social Disadvantage, based on the five-yearly Census run by the Australian Bureau of Statistics; and the instrument used to compile the Dropping Off the Edge (DOTE) report, published episodically since 1999.

The most recent SEIFA index ranks Corio, Norlane, and Whittington in the top-most percentile of most disadvantaged suburbs in Victoria [50]. In the two most recent DOTE reports, published in 2014 and 2021, Corio and Norlane (grouped together as Corio–Norlane for statistical purposes) are ranked in the top quintile of disadvantaged areas nationally, and are described as exhibiting "persistent" and "multilayered" disadvantage [51]. All three suburbs have long been characterized by low household incomes, poor education levels, limited digital inclusion, and high levels of poverty and unemployment.

These three suburbs all share a high proportion of people employed in low-paid service sector jobs, people born outside Australia, cultural and linguistically diverse (CALD) groups, Aboriginal and Torres Strait Islander people, and single-parent families. They also share high concentrations of public housing, renters, and people experiencing housing stress. Public housing makes up 22 percent of dwellings in Norlane, 10.6 percent in Corio and 16.7 per cent in Whittington; the national figure is less than 4 percent [50]. Most local public housing was built in the 1950s–1970s and is increasingly run-down. Figures for domestic violence and other crime are considerably higher than average [52]. Public transport options are limited, and many households lack a private vehicle. In addition to these challenges, Whittington, Corio, and Norlane are also subject to persistent place-stigma, which has negative implications for community morale and residents' prospects.

### 4. Methods

The overall project adopted a four-stage research process addressing nine research questions (Table 1), with the first three stages providing evidence that informed the fourth stage—the systems thinking workshop that this paper reports on. The first three stages consisted of:

1.  Wide, exploratory research and environmental scoping (Environmental Scan, January 2021);
2.  Focused statistical analysis examining COVID-19's impact on Australia, the Geelong region, and the three localities of Corio, Norlane, and Whittington (Data Scan, February 2021); and
3.  Targeted interviews and focus group with key service provider experts in the Corio, Norlane, and Whittington communities (Community Consultations, March 2021).

**Table 1.** Research Stages. Asterisks indicate which data collection methods addressed which research questions.

| Research Question | Data Collection Method | | | |
| --- | --- | --- | --- | --- |
| | Environmental Scan | COVID-19 Impact Study | | STICKE Workshop (Proposed) |
| | | Data Scan | Interviews + Focus Group | |
| 1. What are the social and economic factors that drive outcomes, particularly for communities affected by disadvantage? | * | * | * | * |
| 2. What interventions have been used to address these factors to improve community outcomes (Australia + global)? | * | | | |
| 3. What federal and state policies, programs, and funding opportunities target social and economic development to improve community outcomes? | * | | | |
| 4. What policy gaps could be addressed to improve community outcomes? | * | * | * | * |
| 5. What are the key issues to be addressed in a COVID-19 impact assessment? | | * | | |
| 6. How have "disadvantaged" communities in Geelong dealt with the pandemic? | | * | * | |
| 7. What factors facilitated local economic resilience and affirmed social cohesion? | | * | * | |
| 8. What interventions worked best, and what other approaches could have enhanced residents' experiences? | | * | * | |
| 9. What system-wide actions might be taken to overcome obstacles to addressing social disadvantage in Geelong? | | | | * |

The research questions and study design were developed in consultation with the community stakeholders that would participate in the research. The results of the first three stages are published in detail elsewhere (reference removed for anonymity).

Stage four involved a half-day systems thinking workshop from which causal loop diagrams were built to represent the logic of interview data. Community based participatory group model building techniques were used to review the logic model and use it to develop action plans. We worked with community stakeholders across all phases of the project, because such an approach has been found to enhance rigor in qualitative research through the integration of diverse perspectives and interpretations [53].

In stages one to three of the project, qualitative stakeholder interviews and synthesis of existing literature was used to build an understanding of the impacts of acute socioeconomic disadvantage in place [54,55].

For stage four, data collection and analysis were informed by community based participatory system dynamics and utilized group model building (GMB) [49] to produce a system map in the form of a causal loop diagram (CLD). To this end, the workshop made use of a tool (STICKE) developed in collaboration with the World Health Organization Collaborating Centre for Obesity Prevention to facilitate community knowledge exchange to foster shared understanding of complex problems. Systems thinking workshops using STICKE follow a scripted routine to guide participants through steps to create the CLD. A CLD shows not just the factors but also the ways in which they may be causally related to each other and to spatial disadvantage. The process was carefully structured to take

participants through various exercises which result in a CLD (Figure 2) that represents a consensual view on the system's components, relationships, and boundaries.

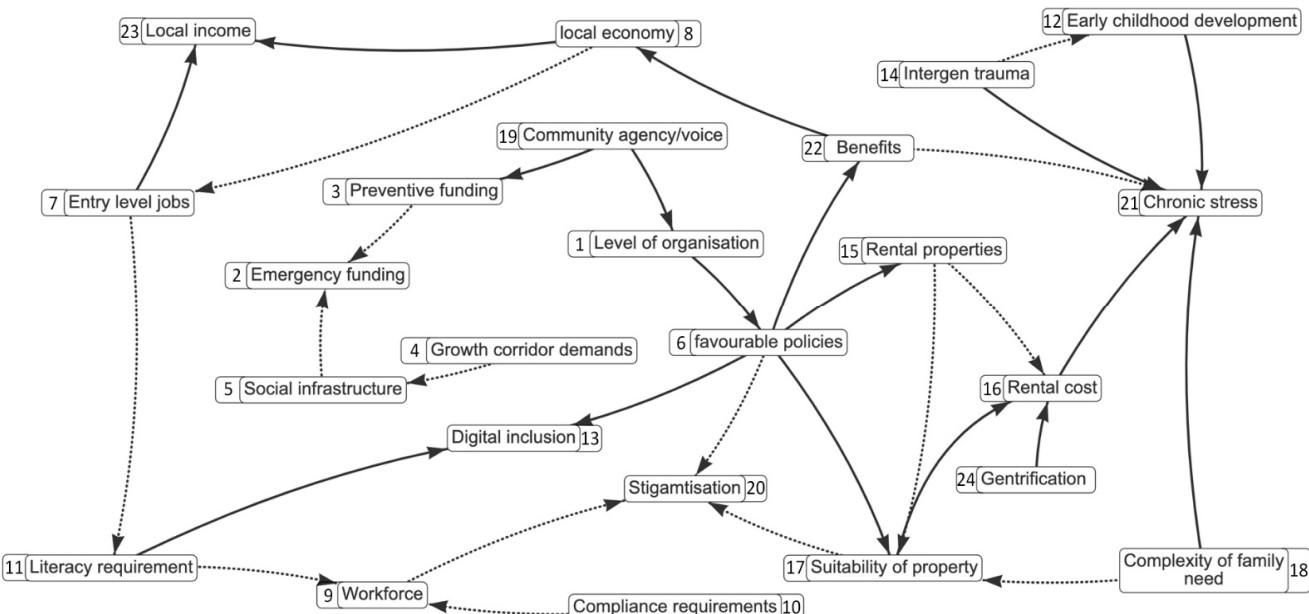

**Figure 2.** Causal loop diagram of the causes and impacts of socioeconomic disadvantage in Whittington, Corio, and Norlane. A solid line denotes a positive causal link between two variables that change in the same direction, while a dotted line indicates a negative causal link between two variables that change in opposite directions.

The intention of these techniques was to surface the mental models of community members regarding the causes of socioeconomic disadvantage in Whittington, Corio, and Norlane, and couple this with the existing evidence base about the causes and common strategies to alleviate the impacts of socioeconomic disadvantage and to develop context specific understanding of potential intervention areas and recommendations for action.

### 4.1. Participants and Recruitment

Stage three: participants in targeted interviews were from key service provider experts in the Corio, Norlane, and Whittington communities, with purposeful sampling to represent each of five practice domains: health and wellbeing, education, housing, employment, and livability. Twelve key community experts whose organizations service the three localities were interviewed. Further details of these participants are given in the stages one to three project report Creative Strategies for Tackling Locational Disadvantage in Geelong (reference removed for anonymity).

Stage four participants included key stakeholders/policy makers in Whittington, Corio, and Norlane and general community members. Purposive sampling was used to recruit participants from health services, education, housing, and employment sectors, and those with expertise and experience in livability issues. The participants included policy actors operating at local (*n* = 6) and state government (*n* = 2) levels, and (again) experts whose organizations service the three localities, including a philanthropic community foundation, a community group that advocates on sustainability issues, a community service provider supporting employment and training for people with disability, and an alliance of organizations that provide youth services. All participants were selected for their in-depth knowledge of the causes and effects of disadvantage across the three communities. Community members were identified through snowball sampling recommended by key stakeholders.

### 4.2. Data Collection and Analysis

Stages one to three: data from the first set of qualitative interviews combined with a synthesis of the literature (the evidence review) were used to generate recommendations published in the stages one to three report (reference removed for anonymity). Three recommendations were made. The first describes an approach for governmental agencies to follow in working with communities in Corio, Norlane, and Whittington on the issues that require priority action. It also describes the roles that key community informants have suggested CoGG can play in this work. The second recommendation identifies 14 priority areas for action. The final recommendation outlines how these priority areas could be addressed through 10 types of initiative. In the interests of obtaining buy-in and action from key stakeholders, the recommendations, together with the findings of the evidence review, were briefly outlined to stage four participants in the introduction to their workshop.

In stage four, we collected data via a qualitative structured group process using video conferencing facilities. The original plan was to conduct these sessions in person, but travel restrictions due to COVID-19 meant we were unable to conduct these face-to-face and so the research team facilitated the sessions remotely. The session was conducted in February, 2022. The session format is described in Table 2.

**Table 2.** Workshop format and data collection for stage four.

| Agenda Item | Time (mins) | Description |
|---|---|---|
| Welcome | 15 | The study lead introduced the session and the purpose of the study, welcomed people to the session, and outlined the workshop structure and aims. |
| Evidence Brief | 15 | Participants were presented with an evidence brief providing the most recent information about spatial disadvantage research. The evidence brief also presented information on what is known about spatial disadvantage in Geelong and how this has changed during the COVID-19 pandemic. |
| Behaviors over time introduction | 20 | A process explaining how the impacts of socioeconomic disadvantage might be mapped over time was described to the participants. |
| Behaviors over time | 20 | Working in three small groups of 5 or 6 people (in breakout rooms), individual participants were then asked to map impacts that they identified of socioeconomic disadvantage. The small groups were preselected by the researchers for a diversity of expertise in each group. The group then prioritized which of these impacts would be shared with the wider group. |
| Model review introduction | 20 | All participants returned to the main room. The process used to develop the maps in STICKE was described to them and the map presented by building the map theme by theme from the impacts prioritized in the previous session. The meaning of the variables, direction, and style of arrows was described to participants. |
| Model review | 30 | The group as a whole was then invited to review the maps of the system relating to the causes and effects of socioeconomic disadvantage in Geelong and identify where they felt something was missing. They were offered the opportunity to augment the maps and add things they felt were missing. This provided an updated map that reflected the individual participant's understanding of the system and provided data on the maps for future review. |
| Action review introduction | 10 | Again in the main room, using their augmented maps participants identified the places on the map where existing action was happening and indicated this by pacing (digitally, using the Zoom stamp function) a red heart on the part of the map the action was affecting, and to consider where more action was needed and to place a black cross on the map where they felt it was required, and to circle the areas of the map (digitally, using the Zoom draw function) where they felt they had power and agency to act to change and reduce the impacts of socioeconomic disadvantage. |
| Action ideas and prioritize | 20 | Using the further developed maps, participants were then asked to consider actions that might be taken to alleviate the impacts of spatial disadvantage. These actions were described on the action ideas template and participants were asked to identify which parts of the map the action would impact. |
| Prioritize | 20 | Working in the same three small groups of 5 or 6 people (in breakout rooms), participants were then asked to share their ideas with each other and prioritize these ideas in order from highest to lowest priority. They were asked to prioritize considering both the feasibility of the action and the likely impact of the action. |
| Group summary to room | 20 | Back in the main room, the small working groups created in the previous step reported their priority actions to the rest of the group and these actions were recorded and displayed. |
| Collate, vote and commit | 15 | Participants were then asked to identify which ideas they would like to pursue if and when further discussions took place about how to implement the ideas. |
| Next steps and close | 10 | The next steps in the project were described and the meeting drawn to a close. |

*4.3. Ethics*

Ethics approval was granted by a university Human Research Ethics Committee (project no. SEBE-2021-02-MOD02). Written informed consent was obtained from all participants.

## 5. Results

After participants were briefed on the overall research question, breakout groups created a set of responses that were then grouped into a causal loop diagram. The diagram included 24 separate variables (Figure 2) under 10 themes (bolded below), which are drawn from the social, economic, health, and urban environment determinants of socioeconomic disadvantage identified in stages one to three: Policy (governmental—for community and social infrastructure), Employment and work, Socioeconomic Position, Education, Personal Health Status, Housing (access to affordable and appropriate housing), Healthcare and Social Services, Social Networks, Support and Exclusion, Crime and Safety, and Socio-demographics.

These 24 variables, which can be considered as the obstacles in the system to addressing socioeconomic disadvantage, consisted of:

**Policy (governmental—for community and social infrastructure)**

1. Lack of coordination between levels of governmental (City, State and Federal) organizations;
2. The necessity and timing of emergency governmental funding; in relation to
3. The need to provide timely preventative governmental funding (usually not enough and often too late);
4. Increasing social infrastructure demands in growth corridors, competing for resources with;
5. Demands for upgrading inadequate social infrastructure in Whittington, Corio, and Norlane;
6. Favorable (and, conversely, non-favorable) broad economic policies—such as negative gearing, the low level of New Start, and other welfare payments—undermining socioeconomic development.

**Employment and work**

7. Availability of entry level jobs with the demise of manufacturing in Geelong and a related decline in
8. The local economy, especially organizations that used to serve manufacturers and neighborhood shopping precincts;
9. Workforce impacts;
10. Compliance requirements (of workers);
11. Literacy requirements (rising demands for literacy, especially digital and numerical for workers).

**Education**

12. Lack of support for early childhood development;
13. Digital inclusion.

**Personal Health Status**

14. Intergenerational trauma.

**Access to affordable and appropriate housing**

15. Availability and quality of rental properties;
16. Price of rental properties;
17. Housing (property) suitability (with more complex clients in the public sector and related impacts on perception of safety and social harmony).

**Healthcare and Social Services**

18. Complexity of family needs.

**Social Networks, Support & Exclusion**

19. Community agency and voice.

**Crime and Safety**

20.　Place stigmatization.

**Health and Wellbeing**

21.　Chronic stress (impacting mental health and wellbeing).

**Socioeconomic Position**

22.　Levels of unemployment (and other) benefits;
23.　Local income levels.

**Socio-demographics**

24.　Gentrification.

In the casual loop diagram, it is worth noting some variables are identifiable as key causes (that is, no arrows point to them), such as Gentrification, Complexity of Family Need, and Compliance Requirements. At the same time, other variables are identifiable as key effects (that is, no arrows emerge from them), such as Local Income and Chronic Stress.

Participants identified 13 separate actions to address socioeconomic disadvantage in Whittington, Corio, and Norlane (Table 3), including five with a governmental funding policy focus, one with a safety focus (in relation to neighborhood reputation and place-stigma), one with a housing focus, one with an employment and work focus, and four with social networks, support and exclusion focus:

- Governmental funding policy actions crossed practice domains and ranged from the improvement of local, state and federal governmental funding policy in relation to the location of services and support for community organizations, and employment incentives, to longer-term planning of strategies to alleviate the impacts of spatial disadvantage, and empowering communities to inform decisions;
- In relation to livability, it was suggested that place-stigma could be improved by changing the language used in policy strategies from that of deficit to a positive framing for communities experiencing disadvantage;
- It was suggested that increasing the supply of affordable and appropriate housing should be stipulated as a prerequisite of all housing development in the three communities;
- In relation to employment, it was suggested that support should be tailored to the needs of local small business owners; and
- Social networks, support and exclusion actions crossed practice domains and included empowering the voices of local communities through building stronger coalitions to increase advocacy, embedding co-design by the community in all aspects of government social procurement policy, and emphasizing partnership and synergy between service providers, support organizations and the community to ensure place-based responses.

**Table 3.** Prioritized action ideas by map theme and participating community.

| Theme | Policy (Governmental—Funding for Community and Social Infrastructure) | Social Networks, Support & Exclusion | Safety (in Relation to Neighborhood Reputation and Place-Stigma) | Housing (Access to Affordable and Appropriate Housing) | Employment and Work |
|---|---|---|---|---|---|
| **Practice Domain** | Cross-Domain | Cross-Domain | Livability | Housing | Employment |
| **Action** | Locate service across all three communities<br>Support head office location of organizations in the VCs<br>Link incentives to local employment, and rental costs for local organizations via the G21 Region Opportunities for Work (GROW) initiative<br>More flexible and recurrent funding tailored to local need<br>Ensure a long-term planning and implementation window for all actions<br>Empower local community to inform the allocation of funding in the VCs | Bring people into the communities through events linked to local renewal<br>Build stronger coalitions to increase community advocacy and community voice<br>Embed co-design by the community in all aspects of government social procurement policy<br>Emphasize partnership and synergy between service providers, support organizations and the community to ensure place-based responses. | Changing the language used in policy strategies from that of deficit to a positive framing for communities experiencing disadvantage | Create suitable and affordable housing as a starting point for building development | Tailor support to the needs of local/ small business owners by targeting new and existing support to them |

## 6. Discussion

Extant literature and the whole notion of a "wicked problem" had indicated that understanding and ameliorating social-spatial disadvantage in multiple localities was going to involve many intersecting causes and effects. However, the STICKE Workshop allowed participants to distil both a common understanding of the key drivers of socio-spatial disadvantage in Corio-Norlane and Whittington and a set of practical actions that could be taken. Further, because those who participated in the systems thinking workshop were members of relevant CoGG departments as well as community service and other local organizations, their convergence of opinion on strategies and priorities bodes well for effective interventions. If one of the key recommendations of the (Stage 3) Creative Strategies for Tackling Locational Disadvantage in Geelong report was to ensure the co-design of interventions, then this workshop was a demonstration of the ease as well as the power and possibilities of this approach.

From a recognition that socio-spatial disadvantage in these three localities was founded on an array of specific government policies and practices (especially around the timing and targeting of funding as well as connection to community priorities), a growing crisis in affordable appropriate housing as well as adequate employment opportunities along with a need to remove particular barriers (around digital and other forms of literacy, early childhood education, intergenerational trauma) and address place stigma, there emerged a set of priority actions. They are noted above. In addition, many of these can indeed be realized at local government level, including:

- Around government funding:
  - Flexible, co-designed, and more recurrent funding streams related to community-defined priorities; and
  - A genuinely broad-based community consultation process to elicit needs and to co-design appropriate policy and service responses.
- In relation to employment:
  - Initiatives like GROW (Geelong Region Opportunities for Work) and social procurement policies can be extended with incentives to local employers to hire locally;
  - Offer bespoke support to local businesses;
  - Council could offer subsidized facilities to not-for-profit businesses; and
  - Continue the process of co-designing and renewing local strip shopping areas.
- To address housing issues: Policies to facilitate diversity in form and ensure that, for example, new social housing is fit for purpose, including for those with special needs, disability and the aged.
- For education: Replicating and extending an existing initiative in Norlane that combines service providers into a multi-faceted response to early childhood (under 5) needs. The suggestion was that this model could be also placed in Corio and Whittington and extended to children under 12.
- To ease place-based stigma:
  - Council support to local festivals and events; and
  - Continue to develop green neighborhoods and shopping precincts.

### 6.1. Strengths

The systems approach followed might usefully support other communities in translating systems theory into system-wide practice. In particular, the creation of visual representations of complexity in the form of a CLD enabled community members to build a shared understanding of community systems as mental modes, and to collaborate in those places where they felt action is possible [56]. This led to the direct sharing of knowledge and experience between people with and without lived experience of disadvantage, enabling diverse stakeholders to generate a mutually agreed plan of action for overcoming city-scale obstacles to change.

Further, the use of co-design principles provided new insights and deeper engagement than more traditional approaches to intervention design. Similarly, the multi-phase approach to the study meant data were able to be considered, synthesized, and fed into subsequent steps, demonstrating to the community the respect held for their data and the utility it had for supporting their efforts to improve the socioeconomic outlook of residents in Corio, Norlane, and Whittington. Moreover, the ability to locate initiatives between systemic and systematic efforts also lends itself to policy development [57], especially when such initiatives are arrived at in consensus with those who inform and implement policy.

At the end of the STICKE workshop, participants were asked if they were willing to be involved in the implementation of the various initiatives. Many were, and as the group comprised local government representatives along with key members and decision-makers in service providers and community groups, there is a very strong likelihood that these commitments will be realized. Therefore, the research process itself has and will lead directly to actions which have been derived from the three critical communities and offer an example of local empowerment. Ultimately such actions should lead to an amelioration of social and spatial disadvantage in these areas, a hugely important outcome for any research project.

In sum, some key tendencies of effective place-based interventions were affirmed: (1) developing meaningful community engagement that is respectful, empowering, sustainable, inclusive, and genuinely participatory; (2) employing co-design approaches that recognize and build on the community's existing knowledge and strengths; (3) involving strong partnerships and relationships of trust; and (4) fostering holistic thinking and adaptability [58,59].

### 6.2. Limitations

It is widely acknowledged that participatory processes are more powerful when conducted in person, and due to COVID the research team and participants were unable to physically attend the workshop. The use of videoconferencing is emerging in this type of research and this project represents one of the first to present a hybrid model of activities in place (via videoconferencing) and data collection and synthesis remotely. While not ideal, it may represent an incremental step in making such methods available for less accessible communities that, prior to high quality videoconferencing facilities, was unavailable.

In the interviews and STICKE workshops the participating key stakeholders were selected purposively, and community members selected through snowball sampling on the basis of key stakeholder recommendation. While this achieved a wide range of stakeholder and community representation, including residents and key community experts whose organizations service the three localities, the potential limitations of this approach should be acknowledged in relation to the risk that particular voices in the community may have been unrepresented. In an ideal world, with a greater timeframe for data collection, and less restricted access to the communities due to COVID limitations, recruitment could better target wider representation of those experiencing and addressing disadvantage.

The final discussion in the STICKE workshop asked participants to reflect on the process and offer suggestions for improvement. While most were deeply impressed and enthusiastic about the process, one very useful suggestion was to ensure that there was a clarification of key terms and concepts at the beginning of the process, to ensure that there was a common understanding and language with shared terminology that could then be deployed. This will be taken up in future workshops.

### 6.3. Implications for Practice

It is clear there is desire and will in Corio, Norlane, and Whittington for change to alleviate the impacts of socioeconomic disadvantage. These efforts also have the potential to positively impact health and wellbeing, education, housing, employment, and livability, and as CoGG was central to the consensus built around the actions that were prioritized, most of these actions should be translatable to policy positions. Moreover, as the actions

are informed and supported by community members, advocates and community support organizations, residents in Corio, Norlane, and Whittington are likely to be receptive to these interventions and working with CoGG and support organizations to develop them.

It is also worth highlighting that most actions for alleviating socioeconomic suggested in the workshop crossed practice domains, which is consistent with what the research suggests are the most effective and commonly used intervention strategies [14,58–62].

### *6.4. Future Research*

Further research is needed that builds on the community engagement and good will generated here to see whether these actions can be translated into practical policy implemented in Corio, Norlane, and Whittington and what effect these may have on concentrations of socioeconomic disadvantage. Previous examples using these techniques in community wide intervention design have proven effective and have become embedded as policy positions in multiple jurisdictions [17,58,63–65], although difficulties have been highlighted in implementing partnership and participation initiatives in such contexts [66]. There is also an opportunity to assess the effectiveness of the community engagement process itself as well as the actions that emerged from it.

### 7. Conclusions

This research had three primary objectives: (1) to create a shared understanding of the impacts of acute socioeconomic disadvantage in place; (2) develop a set of practical ideas to address some of these impacts in Corio, Norlane, and Whittington; and (3) identify which of these ideas should be prioritized. To this end, a co-design workshop was held, using realist inquiry and system mapping, with 18 community stakeholders. In relation to the first objective, twenty-four distinct impacts and causes of spatial disadvantage were identified. In relation to the second, community members developed 13 intervention ideas for action to overcome these obstacles to alleviating disadvantage: five with a governmental funding policy focus, one with a safety focus (in relation to place-stigma), one with a housing focus, one with an employment focus, and four with social networks, support and exclusion focus. In relation to the third objective, there were several very specific suggestions for policy and practice interventions that the City of Greater Geelong can enact, offering a set of long term and more structural as well as shorter term actions. There were also suggestions on how such policies can be enacted, with members of the workshop offering to assist in implementation boding well for an impactful outcome of this research and its strategies for action.

Further, the research confirmed that the most effective and commonly used intervention strategies for alleviating disadvantage cross domains of practice. In other words, place-based interventions ought to be considered holistically, be multi-pronged in focus and draw on cross-sector collaboration.

Lastly, the research underlined the efficacy of systems approaches to support communities in translating systems theory into system-wide practice.

**Author Contributions:** Conceptualization, R.T. and L.J.; methodology, S.A., R.T. and L.J.; formal analysis, R.T. and L.J.; investigation, S.A., R.T., L.J. and J.L.; writing—original draft preparation, R.T., L.J., J.L. and S.A.; writing—review and editing, R.T., L.J., J.L. and S.A.; project administration, R.T.; funding acquisition, R.T., L.J. and J.L. All authors have read and agreed to the published version of the manuscript.

**Funding:** This research was funded by the City of Greater Geelong, no grant number.

**Institutional Review Board Statement:** The study was conducted in accordance with the Declaration of Helsinki and approved by the Deakin University Human Research Ethics Committee (DUHREC) (SEBE-2021-02-MOD02, 1 March 2021).

**Informed Consent Statement:** Informed consent was obtained from all subjects involved in the study.

**Data Availability Statement:** Not applicable.

**Acknowledgments:** We thank the anonymous reviewers for their valuable comments that have helped improve the paper.

**Conflicts of Interest:** The authors declare no conflict of interest.

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
