# Peer review of "Strategies for Alleviating Spatial Disadvantage: A Systems Thinking Analysis and Plan of Action"

_sustainability, doi:10.3390/su141710477_

Round 1

Reviewer 1 Report

In this article, the authors document an innovative process through which 18 community actors collaboratively specified the causes of socioeconomic inequality and spatially concentrated disadvantage in a suburb of Victoria, Australia, and generate proposals for ameliorative interventions. The structured methodology (STICKE) was locally developed at Deakin University. 

Overall, I think the article delivers on its goal, which is to draw lessons from this process for future processes. I am left wondering about several aspects:

First, on page 7 (section 4.1) the authors discuss how the participating key stakeholders were selected purposively and community members through snowball sampling on the basis of key stakeholder recommendation. I would appreciate additional discussion on the potential limitations of this approach. Is there a risk that particular voices in the community may be left out? Also, the focus seems to be on locally situated actors. Given the extensive involvement of the federal and state governments in local governance and social policy, is there a place for involving policy actors operating at those levels? If no, why not?

Regarding the workshop format described on page 8, I would appreciate additional discussion of the following:

·      Is there a risk that the framing of issues in the evidence brief led the participants in particular directions? What efforts were taken to make the brief “neutral”? Did the workshop leaders find that participants introduced issues or facts not present in the evidence brief?

·      Did the model review and action review segments involve the whole group, or subgroups? It would be helpful to clarify exactly how the groups were broken up at each phase of the workshop.

On page 10, Figure 2 displays different line styles but it is not immediately clear what these mean. Can these be described in the caption? Also, some nodes in the diagram appear to be ultimate causes (that is, no arrows point to them), such as Gentrification, Complexity of Family Need, and Compliance Requirements. At the same time, I believe only one node appears to be an ultimate effect (that is, no arrows emerge from it): Local Income. Are we to take from this that (personal or household) income is the core operationalization of disadvantage? Perhaps one solution to this reader’s confusion would be to more directly link the list of 24 items on pages 10 and 11 to the causal loop diagram by (a) turning the list into a table with the numbered items coded by whether they are causes, intervening factors, or effects, and (b) numbering the items in the diagram so that the reader can identify the corresponding item in the list/table.

One page 11, Table 2, the two “Cross-domain” column headers are confusing to the reader. Perhaps the “theme” row should be at the top and the “practice” row below. 

Finally, I hope that the authors will correct the several spelling errors in the text and figures.

Author Response

Dear Reviewer,

Thank you for your comments, please find our responses to your comments attached. 

Kind regards,

Jian Liang

Reviewer 2 Report

Dear authors,

First I would like to congratulate you for the research theme and for your work.  

After a careful reading, I have the following recommendations:

4. Methods

a). As the research consists in a series of methods, I suggest a scheme that would highlight the 4 stages and the connection/interdependencies between them.

b). For the qualitative research, before Table 1 the research scope and objectives should be specified.

c). I suggest a more detailed description of the participants (4.1.) - how the selection was made, what were the expected outcomes from the participants, what categories of participants did you consider, etc.

7. Conclusions

a). I suggest to compare the results of your research with other similar situations, to try to validate/invalidate the results of your research. I appreciate that in this way you can emphasize both the scientific contribution (the results of your research) as well as the practical implications of your work.

b) I suggest to write the conclusions based on the research objectives.

Author Response

Dear Reviewer,

Thank you for your comments, please find our responses in the file attached. Thank you.

Kind regards,

Jian Liang

Reviewer 3 Report

This study is focused on the impacts and solutions of socioeconomic disadvantage. Although previous research on this topic is extensive, systematic studies are few. What is more, research on solutions to socio-spatial disadvantage is rare. This study is an essential supplement to the research field of social disadvantage.

Author Response

Dear reviewer, 

Thank you for your comments, we have conducted spell check carefully as you suggested. Thank you!

Kind regards,

Jian Liang